# Apixaban Inhibits Progression of Experimental Diabetic Nephropathy by Blocking Advanced Glycation End Product-Receptor Axis

**DOI:** 10.3390/ijms26073007

**Published:** 2025-03-26

**Authors:** Takanori Matsui, Ami Sotokawauchi, Yuri Nishino, Yoshinori Koga, Sho-ichi Yamagishi

**Affiliations:** 1Department of Bioscience and Biotechnology, Fukui Prefectural University, Eiheiji 910-1195, Japan; matsuita@fpu.ac.jp; 2Department of Pediatric Surgery, Kurume University School of Medicine, Kurume 830-0011, Japan; sotokawauchi_ami@med.kurume-u.ac.jp (A.S.); koga_yoshinori@med.kurume-u.ac.jp (Y.K.); 3Department of Medicine, Division of Nephrology, Kurume University School of Medicine, Kurume 830-0011, Japan; nishino_yuri@med.kurume-u.ac.jp; 4Department of Medicine, Division of Diabetes, Metabolism, and Endocrinology, Showa University Graduate School of Medicine, Tokyo 142-8666, Japan

**Keywords:** AGEs, RAGE, diabetic nephropathy, apixaban

## Abstract

Diabetes is associated with an increased risk of thromboembolism. However, the effects of apixaban, a factor Xa inhibitor on diabetic nephropathy, remain unknown. Six-week-old Wistar rats received a single 60 mg/kg intraperitoneal injection of streptozotocin to produce a model of type 1 diabetes. Type 1 diabetic and non-diabetic control rats were treated with or without apixaban orally for 8 weeks, and blood and kidneys were obtained for biochemical, real-time reverse transcription-polymerase chain reaction (RT-PCR) and morphological analyses. Although apixaban treatment did not affect glycemic or lipid parameters, it significantly (*p* < 0.01) inhibited the increases in advanced glycation end products (AGEs), the receptor for AGEs (RAGE) mRNA and protein levels, 8-hydroxy-2′-deoxyguanosine (8-OHdG), and NADPH oxidase-driven superoxide generation in diabetic rats at 14 weeks old. Compared with non-diabetic rats, gene and protein expression levels of monocyte chemoattractant protein-1 (MCP-1), vascular cell adhesion molecule-1 (VCAM-1), transforming growth factor-β (TGF-β), connective tissue growth factor (CTGF), and fibronectin were increased in 14-week-old diabetic rats, which were associated with enhanced renal expression of kidney injury molecule-1 (KIM-1) and Mac-3, increased extracellular matrix accumulation in the kidneys, and elevated urinary excretion levels of protein and KIM-1, all of which were significantly inhibited by the treatment with apixaban. Urine KIM-1 levels were significantly (*p* < 0.01) and positively correlated with AGEs (*r* = 0.690) and 8-OHdG (*r* = 0.793) in the kidneys and serum 8-OHdG levels (*r* = 0.823). Our present findings suggest that apixaban could ameliorate renal injury in streptozotocin-induced type 1 diabetic rats partly by blocking the AGE-RAGE-oxidative stress axis in diabetic kidneys.

## 1. Introduction

Diabetic nephropathy is a leading cause of end-stage renal disease and is associated with the increased risk of cardiovascular events in patients with diabetes, which have accounted for high mortality rates in these patients [1,2]. Various metabolic and hemodynamic pathways are activated under diabetic conditions; chronic hyperglycemia-induced oxidative stress generation, inflammation, and advanced glycation end products (AGEs) formation have been reported to play a central role in the development and progression of diabetic nephropathy, partly by inducing thrombotic and fibrotic reactions, as well as activation of the renin–angiotensin–aldosterone system [3,4,5,6]. Randomized clinical trials have shown that inhibitors of sodium–glucose cotransporter 2 or the renin–angiotensin system, a glucagon-like peptide-1 receptor agonist, semaglutide, and a non-steroidal mineralocorticoid receptor antagonist, finerenone, significantly reduce the risk of renal composite outcomes in diabetic patients [7,8,9,10]. However, there is a high residual risk of chronic kidney disease progression with current therapies, and a substantial number of diabetic patients still experience end-stage renal disease [1,2]. Therefore, a novel therapeutic target should be clarified for the treatment of diabetic nephropathy.

Atrial fibrillation (AF) is the most common disorder of cardiac rhythm, and the number of patients with AF is increasing all over the world, especially in an aging society [1,2,3,4]. Most of the AF patients are affected by various comorbidities, such as hypertension, diabetes, obesity, and chronic kidney disease, all of which could increase the risk of AF-associated complications, including stroke, heart failure, thromboembolic events, and death [11,12,13,14,15,16,17,18,19,20]. A systematic review of direct oral anticoagulants (DOACs) in non-valvular AF patients with moderate chronic kidney disease revealed that dabigatran 150 mg twice daily and an inhibitor of factor Xa, apixaban were superior to warfarin for stroke and systemic embolic events prevention, while apixaban and edoxaban significantly reduced the risk of major hemorrhage more than warfarin [21,22]. Moreover, apixaban was more effective than warfarin in reducing the risk of stroke and death, irrespective of renal function [23]. In addition, secondary analyses of randomized clinical trials with DOACs revealed that there were no treatment interactions with regard to the prevention of systemic embolic events or stroke for any of the DOACs in patients with chronic kidney disease, diabetes, or old age [24]. Since AF and chronic kidney disease patients share risk factors, such as diabetes [11,12,13,14,15,16,17,18,19,20], the above-mentioned findings suggest that apixaban may be preferable to warfarin as a therapeutic option of anticoagulants for the prophylaxis of stroke in non-valvular AF patients with diabetic nephropathy. Furthermore, given the pathological role of oxidative stress- and/or AGE-evoked hypercoagulability in diabetic nephropathy [3,4,5,6], apixaban may exert beneficial effects on diabetic nephropathy. However, the clinical efficacy of apixaban in diabetic nephropathy remains to be elucidated. This is a rationale of apixaban selection in the present study for investigating the protective role of factor Xa inhibitors against diabetic nephropathy.

Non-enzymatic modification of monosaccharides, such as glucose, glyceraldehyde, and fructose, react with amino groups of proteins, lipids, and nucleic acids to form senescent macromolecules called AGEs, whose formation and accumulation have been shown to progress under a normal aging process, especially under hyperglycemic or inflammatory conditions, including diabetes, chronic kidney disease, and obesity [25,26,27,28,29,30,31]. Accumulating evidence has suggested that tertiary structural alteration of biomolecules modified by AGEs and their interaction with a receptor RAGE play a pathological role in the development and progression of various diabetes- or aging-related disorders, including diabetic nephropathy, AF, atherosclerotic cardiovascular disease, heart failure, and osteoporosis [19,25,26,27,28,29,30,31]. Given the bidirectional relationship between chronic kidney disease progression and cardiovascular events, including AF and heart failure in diabetic patients [11,12,13,14,19,32], the AGE-RAGE axis could be a therapeutic target for diabetes- or aging-related disorders, such as diabetic nephropathy in patients complicated with AF. However, the effects of apixaban on the AGE-RAGE axis in diabetic kidneys remain unexplored. In this study, we examined for the first time whether apixaban could inhibit the progression of diabetic nephropathy by reducing the AGE-RAGE axis in streptozotocin-induced type 1 diabetic rats. The novelty of this study is to clarify whether and how apixaban, one of the widely used DOACs for the treatment of non-valvular AF, could inhibit the progression of diabetic nephropathy, which is commonly associated with this most prevalent cardiac arrhythmia.

## 2. Results

### 2.1. General Characteristics of Experimental Animals

General characteristics of experimental animals are shown in Table 1. Compared with control rats, body weight (BW), heart rate, and diastolic blood pressure (BP) were significantly lower in streptozotocin-induced type 1 diabetic rats, while blood glucose (BG), glycated hemoglobin (HbA1c), total cholesterol, triglycerides, high-density lipoprotein (HDL)-cholesterol, and blood urea nitrogen (BUN) were elevated. There were no significant differences in these clinical variables between type 1 diabetic rats with or without apixaban treatment.

### 2.2. Effects of Apixaban on AGE-RAGE-Oxidative Stress Axis in the Kidneys of Streptozotocin-Induced Type 1 Diabetic Rats

As shown in Figure 1 and Table 1, although apixaban treatment for 8 weeks did not affect markers of AGE-RAGE-oxidative stress, including gene expression levels of NADPH oxidase components in non-diabetic control rats at 14 weeks of age, it significantly inhibited the increases in AGEs, RAGE mRNA and protein levels, 8-hydroxy-2′-deoxyguanosine (8-OHdG), an oxidative stress marker, NADPH oxidase-driven superoxide generation, and gene expression levels of components of NADPH oxidase, such as *Nox1*, *Nox2*, *Nox4*, *p22phox*, and *p47phox* in the kidneys of 14-week-old diabetic rats in association with the reduction of serum 8-OHdG levels.

### 2.3. Effects of Apixaban on Inflammatory Reactions in the Kidneys of Streptozotocin-Induced Diabetic Rats

As shown in Figure 2, mRNA and protein levels of monocyte chemoattractant protein-1 (MCP-1) and vascular cell adhesion molecule-1 (VCAM-1) and expression of Mac-3, a marker of macrophages in the kidneys of 14-week-old streptozotocin-induced diabetic rats, were significantly higher than those of non-diabetic control rats of the same age, all of which were significantly inhibited by the treatment with apixaban for 8 weeks.

### 2.4. Effects of Apixaban on Fibrotic Reactions in the Kidneys of Streptozotocin-Induced Diabetic Rats

As shown in Figure 3, renal gene and protein expression levels of transforming growth factor-β (TGF-β), connective tissue growth factor (CTGF), and fibronectin were increased in 14-week-old streptozotocin-induced diabetic rats, which were associated with enhanced extracellular matrix accumulation in the diabetic kidneys evaluated by Masson’s trichrome staining. Apixaban treatment for 8 weeks significantly inhibited these fibrotic changes in the kidneys of diabetic rats at 14 weeks of age. Although serum creatinine or BUN levels did not differ between apixaban-treated and non-treated type 1 diabetic rats, apixaban treatment for 8 weeks significantly reduced proteinuria, urinary excretion levels of kidney injury molecule-1 (KIM-1) and its renal expression in 14-week-old diabetic rats (Figure 3 and Table 1). There was a significant positive correlation of urinary KIM-1 levels with renal AGEs as well as renal and serum 8-OHdG levels (Figure 3i–k).

### 2.5. Effects of Apixaban on Protease-Activated Receptor-1 (PAR-1) and Protease-Activated Receptor-2 (PAR-2) Protein and mRNA Levels in the Kidneys of Streptozotocin-Induced Diabetic Rats

Moreover, gene and protein expression levels of PAR-1 and PAR-2 were increased in diabetic rats at 14 weeks of age compared with non-diabetic control rats at the same age, both of which were inhibited by the treatment of apixaban for 8 weeks (Figure 4).

## 3. Discussion

Diabetes is regarded as a prothrombotic state, which is characterized by endothelial dysfunction, platelet activation, hypercoagulability, and impaired fibrinolysis, thereby contributing to the increased risk of vascular complications of diabetes as well as venous thrombosis [33,34,35]. Since DOACs are a recommended therapy for non-valvular AF patients with coronary risk factors, such as diabetes, and that diabetes, chronic kidney disease, and AF are interrelated with each other [11,12,13,14,15,16,17,18,19,20,36], we examined here the effects of an inhibitor of factor Xa apixaban, one of the DOACs on experimental diabetic nephropathy.

We found in this study for the first time that (1) treatment with apixaban for 8 weeks significantly inhibited the increases in AGEs, RAGE gene and protein expression, an oxidative stress marker 8-OHdG levels, NADPH oxidase-driven superoxide generation, gene expression levels of components of NADPH oxidase except for p67phox in the kidneys of 14-week-old streptozotocin-induced type 1 diabetic rats in association with the reduction of serum 8-OHdG values, (2) gene and protein expression of MCP-1 and VCAM-1, chemokine and leucocyte cell adhesion molecule, respectively, and macrophage infiltration into the kidneys were significantly higher in 14-week old diabetic rats than non-diabetic rats of the same age, all of which were improved by apixaban treatment for 8 weeks, (3) mRNA and protein expression levels of TGF-β, CTGF, and fibronectin and extracellular matrix accumulation in the diabetic kidneys were also increased compared with those in non-diabetic control rats, all of which were also inhibited by the treatment with apixaban for 8 weeks, (4) apixaban treatment significantly reduced renal KIM-1 expression, urinary KIM-1 excretion levels, and proteinuria in type 1 diabetic rats at 14 weeks old, (5) urinary KIM-1 levels were positively correlated with kidney AGEs, and renal and serum 8-OHdG levels, and (6) PAR-1 and PAR-2 gene and protein expression levels in the kidneys of 14-week old diabetic rats were increased compared with non-diabetic rats at the same age, both of which were reduced by the treatment with apixaban. Given the accumulating evidence to show that blockade of the AGE-RAGE-oxidative stress axis could inhibit the development and progression of experimental diabetic nephropathy by attenuating inflammatory and fibrotic reactions in the diabetic kidneys [26,34,37,38,39,40,41], our present study suggests that apixaban could inhibit renal damage in streptozotocin-induced type 1 diabetic rats partly by suppressing the AGE-RAGE-mediated NADPH oxidase-derived superoxide generation in the kidneys.

We have previously shown that (1) factor Xa-mediated thrombin formation increases RAGE and MCP-1 gene expression in, and monocytes cell adhesion to, endothelial cells through NADPH oxidase-derived oxidative stress generation via the interaction with PAR-1, (2) AGEs potentiate these harmful effects on endothelial cells by increasing the PAR-1 expression, (3) apixaban inhibits thrombin-PAR-1-mediated superoxide generation and up-regulation of MCP-1 mRNA levels in, and monocytes cell adhesion to, cultured human mesangial cells, and (4) factor Xa evokes superoxide generation and MCP-1 gene expression in renal proximal tubular cells through the interaction with PAR-2, whose expression is augmented by AGEs [42,43,44]. AGE-RAGE-mediated NADPH oxidase-derived oxidative stress has been shown to increase the generation of AGEs and RAGE overexpression in renal intrinsic cells, such as mesangial cells, proximal tubular cells, and endothelial cells [34,37,39,45,46,47]. Thrombin is an agonist of PAR-1, but it cannot activate PAR-2, while factor Xa activates both PAR-1 and PAR-2 [42,43]. These observations suggest that factor Xa- and its derived thrombin-induced oxidative stress generation via the interaction with PAR-2 and PAR-1, respectively, in the kidneys is a molecular target of apixaban; apixaban may inhibit the AGE-RAGE axis in the diabetic kidneys by reducing the NADPH oxidase-derived superoxide generation. In the present study, PAR-1 and PAR-2 mRNA and protein levels were up-regulated in the diabetic kidneys, both of which were significantly reduced by the treatment of apixaban. PAR-1 and PAR-2 expressions were reported to be increased in diabetic kidneys, and dual blockade of PAR-1 and PAR-2 has been shown to improve diabetic nephropathy in type 1 diabetic Akita mice heterozygous for the endothelial nitric oxide synthase gene [48]. These findings further support the concept that there is a pathological crosstalk between the thrombin-PAR-1/factor Xa-PAR-2 system and the AGE-RAGE-oxidative stress axis in experimental diabetic nephropathy. Apixaban could inhibit the crosstalk of the AGE-RAGE axis with the thrombin-PAR-1/factor Xa-PAR-2 system in the diabetic kidneys, thereby suppressing the progression of experimental diabetic nephropathy.

In this study, we found that macrophage infiltration into, and extracellular matrix accumulation in, the diabetic kidneys were inhibited by the treatment with apixaban in association with the reduction of renal and urinary excretion levels of KIM-1, a marker of kidney injury. Since inflammatory and fibrotic reactions, including MCP-1, VCAM-1, TGF-β, CTGF, and fibronectin induction, have been reported to be evoked by oxidative stress generation via the activation of the AGE-RAGE axis [26,34,37,38,39,40,41], the anti-inflammatory, anti-fibrotic, and nephroprotective properties of apixaban observed here may be ascribed partly to its inhibitory effects on the AGE-RAGE oxidative stress axis in the diabetic kidneys. The positive and significant correlation between urinary KIM-1 levels and kidney AGEs, and renal and serum 8-OHdG levels may further support the relevance of the AGE-RAGE oxidative stress axis in the pathogenesis of diabetic nephropathy.

There are several limitations in this study. First, we did not know the dose-response effects of apixaban on diabetic nephropathy in our animals. Second, there was no significant difference in creatinine levels between control and type 1 diabetic rats. Since BW was significantly lower in streptozotocin-treated diabetic rats than control rats, skeletal muscle loss in the former group may partly explain the result. Third, we did not know the exact reason why HDL-cholesterol levels were significantly higher in type 1 diabetic rats than in controls. In any case, further clinical studies are needed to clarify whether a therapeutic dose of apixaban could attenuate renal injury evaluated by urinary excretion levels of KIM-1 and protein in patients with diabetic nephropathy via the suppression of the AGE-RAGE-oxidative stress axis.

## 4. Materials and Methods

### 4.1. Animal Experiments

Apixaban was generously provided by Bristol-Myers Squibb (Lawrenceville, NJ, USA). Male Wistar rats (Crlj:WI) were purchased from Charles River Laboratories Japan (Yokohama, Japan). Six-week-old Wistar rats received a single 60 mg/kg intraperitoneal injection of streptozotocin (Sigma-Aldrich, Saint Louis, MO, USA) dissolved in 10 mM citrate buffer (pH 4.5) as described previously [37]. Non-diabetic control rats received citrate buffer (pH 4.5) alone. Animals with blood glucose levels greater than 200 mg/dL 48 h later were considered diabetic. Diabetic and non-diabetic rats received 5 mg/kg BW apixaban orally once a day according to the preclinical data on pharmacokinetics and pharmacodynamics of apixaban [49]. Diabetic rats received a low-dose of insulin (1 unit of insulin, Sigma-Aldrich) to prevent ketoacidosis, while control rats received vehicle subcutaneously twice a week.

BP and heart rate of 14-week-old rats were measured as described previously [37]. Then, rats were sacrificed after an overnight fasting; blood and kidneys were obtained for biochemical, real-time reverse transcription-polymerase chain reaction (RT-PCR), immunohistochemical, and morphological analyses. Serum 8-OHdG was measured with an enzyme-linked immunosorbent assay (ELISA) kit (KOG-HS10/E; Nikken Seil, Shizuoka, Japan) with inter- and intra-assay coefficients of variation of 1.9–7.1% and 1.0–2.1%, respectively. Absorbance was measured at 450 nm with an ARVO X3 plate reader (PerkinElmer Japan, Yokohama, Japan). Other blood chemistry was analyzed with standard enzymatic methods as described previously [37]. In brief, BG was measured by a method using the glucose oxidase with the Nipro Stat Strip XP3 (Nipro, Osaka, Japan). HbA1c was measured using the enzymatic method by SRL (Tokyo, Japan). T-Chol and HDL-C were enzymatically determined by cholesterol esterase in combination with cholesterol oxidase with the Fuji DRI-CHEM 3000 (Fujifilm, Tokyo, Japan). TG, BUN, and Cre were measured using lipoprotein lipase, urease, and creatinine deiminase-based methods, respectively. Proteinuria was measured with a Protein Assay BCA kit from Nacalai Tesque (Kyoto, Japan) [50].

All experimental procedures were conducted in accordance with the National Institutes Health Guide for Care and Use of Laboratory Animals and were approved by the ethical committee of Kurume University School of Medicine (Approval No. 2017-204, 2018-194).

### 4.2. Measurement of Urinary KIM-1

Urinary KIM-1 excretion was measured with an ELISA kit (ab223858; Abcam, Cambridge, UK) with inter- and intra-assay coefficients of variation of 4.3% and 2.4%, respectively. Absorbance was measured at 450 nm with an ARVO X3 plate reader. Urinary creatinine levels were measured using a Serotec CRE-L (Serotec, Sapporo, Japan).

### 4.3. Measurement of NADPH Oxidase Activity

NADPH oxidase activity in the kidneys was measured by a luminescence assay based on the lucigenin-enhanced chemiluminescence method as described previously [37]. In brief, the kidneys were homogenized in 20 mM HEPES (2-(4-(2-hydroxyethyl)piperazin-1-yl)ethanesulfonic acid) buffer (pH 7.0) containing 100 mM KCl and 1 mM ethylenediaminetetraacetic acid. The homogenate was added to 50 mM sodium phosphate buffer (pH 7.0) containing 1 mM ethylene glycol-bis(β-aminoethyl ether)-N,N,N′,N′-tetraacetic acid and 150 mM sucrose, and then chemiluminescence was measured in the presence of 5 μM lucigenin and 100 μM NADPH.

### 4.4. Immunostaining and Morphological Analysis

Fixation, dehydration, paraffin embedding, and immunostaining of the kidneys were described previously [37]. Antibodies raised against AGEs, RAGE (SC-5563; Santa Cruz Biotechnology, Dallas, TX, USA), 8-OHdG (MOG-020P; Nikken Seil, Shizuoka, Japan), MCP-1 (ab7202; Abcam, Cambridge, UK), VCAM-1 (SC-8304; Santa Cruz Biotechnology), Mac-3 (SC-5571; Santa Cruz Biotechnology), TGF-β (ab92486; Abcam), CTGF (ab6992; Abcam), fibronectin (ab2413; Abcam), KIM-1 (ab47635; Abcam), PAR-1 (bs-2334R; Bioss, Woburn, MA, USA), PAR-2 (SC-13504; Santa Cruz Biotechnology) were used for immunostaining. Immunohistoreactivity in 10 different fields in each sample was measured, and 3-micrometer paraffin sections were stained with Masson’s trichrome for light microscopic analysis [37].

### 4.5. RT-PCR

Real-time RT-PCR was performed as described previously [37]. Identifications of primers for rat RAGE, NADPH oxidase 1 (Nox1), NADPH oxidase 2 (Nox2, also known as gp91phox), NADPH oxidase 4 (Nox4), p22phox, p47phox, p67phox, MCP-1, VCAM-1, CTGF, TGF-β, fibronectin, and 18s rRNA genes were Rn00584249_m1, Rn00586652_m1, Rn00576710_m1, Rn00585380_m1, Rn00577357_m1, Rn00586945_m1, Rn01759078_m1, Rn00580555_m1, Rn00563627_m1, Rn00573960_g1, Rn00572010_m1, Rn00569575_m, and Hs99999901_s1, respectively. Data were normalized by the intensity of 18s-derived signals and then related to the value of control rats.

### 4.6. Statistical Analysis

All values were presented as mean ± standard deviation. Statistical analysis was conducted using R version 4.4.2 software (The R Foundation for Statistical Computing Platform, Vienna, Austria). Normality of the data was tested using the Shapiro–Wilk normality test. ANOVA with the Tukey–Kramer test was performed for normally distributed data, and the Steel–Dwass test was performed for data that did not follow a normal distribution. Correlation between urinary excretion levels of KIM-1 and kidney AGEs, kidney 8-OHdG, or serum 8-OHdG levels was analyzed by Pearson’s correlation test. All *p*-values < 0.05 were considered significant.

## 5. Conclusions

Our present findings suggest that apixaban could ameliorate renal injury in streptozotocin-induced type 1 diabetic rats, partly by blocking the crosstalk between the AGE-RAGE-oxidative stress axis and the thrombin-PAR-1/factor Xa-PAR-2 system in diabetic kidneys. Given the pathological involvement of hypercoagulability in diabetic nephropathy [33,34,35] and the association of diabetes with the increased risk of non-valvular AF [11,19,20], apixaban, one of the therapeutic options for preventing stroke in patients with non-valvular AF, may provide a novel strategy for attenuating renal damage in diabetic patients.

## Figures and Tables

**Figure 1 ijms-26-03007-f001:**
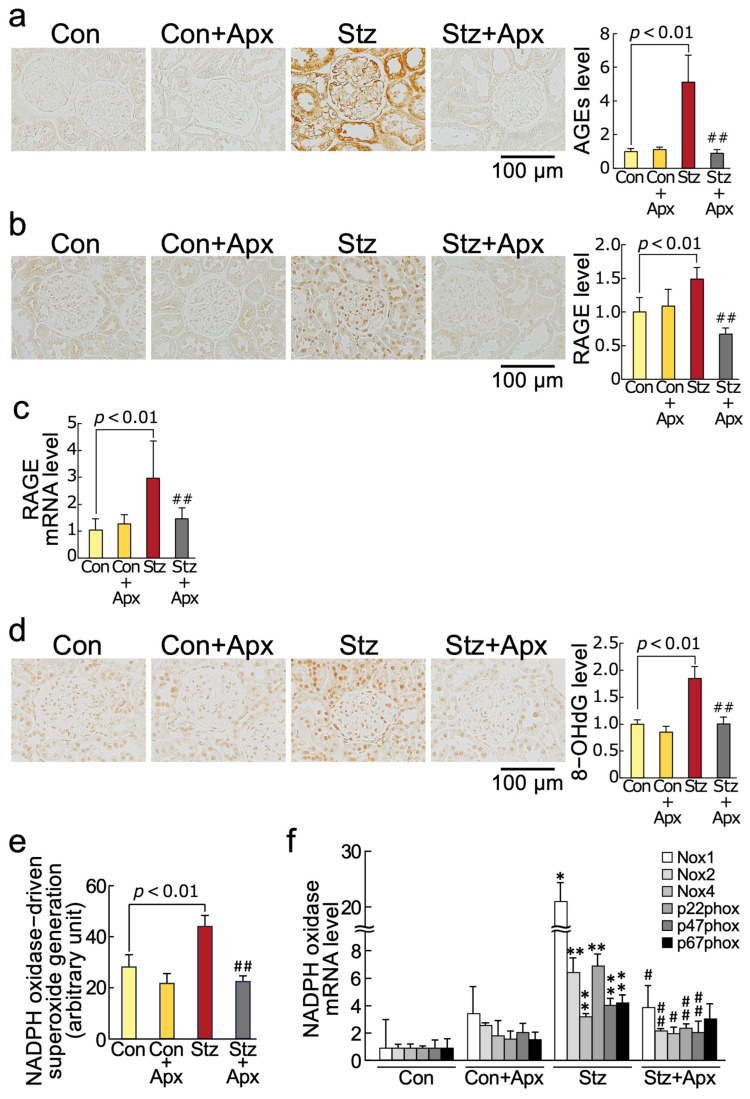
Effects of apixaban (Apx) on AGE-RAGE-oxidative stress axis in the kidneys of streptozotocin-induced diabetic rats. Levels of AGEs (**a**), RAGE protein (**b**), RAGE mRNA (**c**), 8-OHdG (**d**), NADPH oxidase-driven superoxide generation (**e**), and gene expression levels of components of NADPH oxidase (**f**). (**a**,**b**,**d**) Each left panel shows representative immunostainings of AGEs (**a**), RAGE protein (**b**), and 8-OHdG (**d**) in the kidneys. Each right panel shows the quantitative data. * and **, *p* < 0.05 and *p* < 0.01 compared with 14-week-old non-diabetic rats (Con), respectively. # and ##, *p* < 0.05 and *p* < 0.01 compared with 14-week-old diabetic rats (Stz), respectively. AGEs: advanced glycation end products, RAGE: receptor for AGEs, 8-OHdG: 8-hydroxy-2′-deoxyguanosine. Con: N = 6, Con + Apx: N = 5, Stz: N = 5, Stz+Apx: N = 4. Data except for Nox1 mRNA (**f**) were analyzed by ANOVA with Tukey–Kramer test. Data of Nox1 mRNA (**f**) were analyzed by ANOVA with Steel–Dwass test.

**Figure 2 ijms-26-03007-f002:**
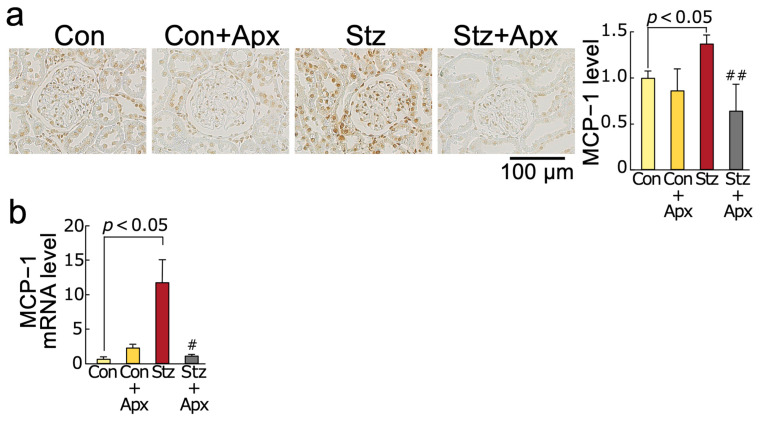
Effects of apixaban (Apx) on inflammatory reactions in the kidneys of streptozotocin-induced diabetic rats. Levels of MCP-1 protein (**a**), MCP-1 mRNA (**b**), VCAM-1 protein (**c**), VCAM-1 mRNA (**d**), and Mac-3 protein (**e**). (**a**,**c**,**e**) Each left panel shows representative immunostainings of MCP-1 protein (**a**), VCAM-1 protein (**c**), and Mac-3 (**e**) in the kidneys. Each right panel shows the quantitative data. # and ##, *p* < 0.05 and *p* < 0.01 compared with 14-week-old diabetic rats (Stz), respectively. Con: 14-week-old non-diabetic rats. MCP-1: monocyte chemoattractant protein-1, VCAM-1: vascular cell adhesion molecule-1. Con: N = 6, Con + Apx: N = 5, Stz: N = 5, Stz + Apx: N = 4. Data except for MCP-1 mRNA (**b**) and VCAM-1 mRNA (**d**) were analyzed by ANOVA with Tukey–Kramer test. Data of MCP-1 mRNA (**b**) and VCAM-1 mRNA (**d**) were analyzed by ANOVA with Steel–Dwass test.

**Figure 3 ijms-26-03007-f003:**
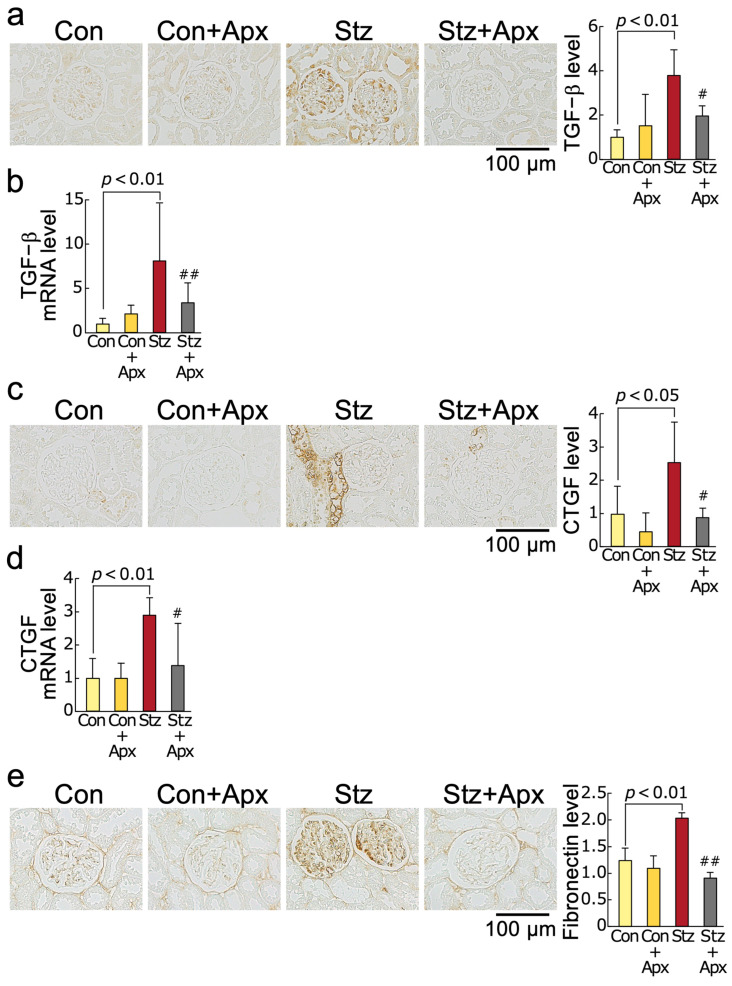
Effects of apixaban (Apx) on fibrotic reactions in the kidneys of streptozotocin-induced diabetic rats. Levels of TGF-β protein (**a**), TGF-β mRNA (**b**), CTGF protein (**c**), CTGF mRNA (**d**), fibronectin protein (**e**), fibronectin mRNA (**f**), glomerular extracellular matrix accumulation (**g**), and KIM-1 protein (**h**). (**a**,**c**,**e**,**g**,**h**) Each left panel shows representative immunostainings of TGF-β protein (**a**), CTGF protein (**c**), fibronectin protein (**e**), and KIM-1 protein (**h**), and glomerular extracellular matrix accumulation (**g**) in the kidneys. (**g**) Masson’s trichrome-stained sections. Each right panel shows the quantitative data. Correlation of urinary excretion levels of KIM-1 with kidney AGEs (**i**), kidney 8-OHdG (**j**), and serum 8-OHdG levels (**k**). # and ##, *p* < 0.05 and *p* < 0.01 compared with 14-week-old diabetic rats (Stz), respectively. Con: 14-week-old non-diabetic rats, TGF-β: transforming growth factor-β, CTGF: connective tissue growth factor, ECM: extracellular matrix, KIM-1: kidney injury molecule-1, AGEs: advanced glycation end products, 8-OHdG: 8-hydroxy-2′-deoxyguanosine. Con: N = 6, Con + Apx: N = 5, Stz: N = 5, Stz + Apx: N = 4. Data (**a**–**h**) were analyzed by ANOVA with Tukey–Kramer test. Data (**i**–**k**) were analyzed by Pearson’s correlation test.

**Figure 4 ijms-26-03007-f004:**
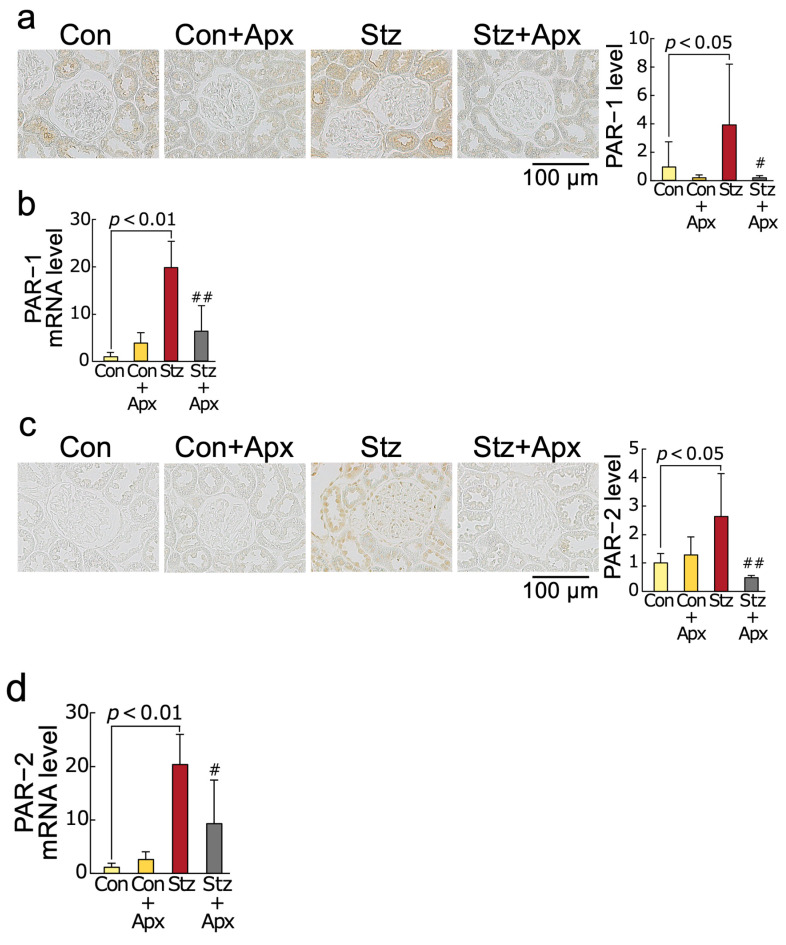
Effects of apixaban (Apx) on PAR-1 and PAR-2 protein and mRNA levels in the kidneys of streptozotocin-induced diabetic rats. Levels of PAR-1 protein (**a**), PAR-1 mRNA (**b**), PAR-2 protein (**c**), and PAR-2 mRNA (**d**). (**a**,**c**) Each left panel shows representative immunostainings of PAR-1 protein (**a**) and PAR-2 protein (**c**) in the kidneys. Each right panel shows the quantitative data. # and ##, *p* < 0.05 and *p* < 0.01 compared with 14-week-old diabetic rats (Stz), respectively. Con: 14-week-old non-diabetic rats, PAR-1: protease-activated receptor-1, PAR-2: protease-activated receptor-2. Con: N = 6, Con + Apx: N = 5, Stz: N = 5, Stz + Apx: N = 4. Data were analyzed by ANOVA with Tukey–Kramer test.

**Table 1 ijms-26-03007-t001:** General characteristics of experimental animals.

	Control	Control + Apixaban	Streptozotocin	Streptozotocin + Apixaban
Number	6	5	5	4
Body weight (g)	467 ± 26	444 ± 30	189 ± 19 **	175 ± 59
HR (beats/min)	361 ± 31	366 ± 16	276 ± 21 **	263 ± 32
SBP (mmHg)	119 ± 10	115 ± 20	115 ± 8	102 ± 11
DBP (mmHg)	93 ± 6	92 ± 15	62 ± 11 **	56 ± 13
BG (mg/dL)	147 ± 37	134 ± 34	573 ± 98 **	666 ± 179
HbA1c (%)	6.1 ± 0.3	5.9 ± 0.3	10.4 ± 0.7 **	11.2 ± 0.5
T-Chol (mg/dL)	63 ± 13	63 ± 6	314 ± 133 **	239 ± 106
TG (mg/dL)	165 ± 52	121 ± 42	1132 ± 721 **	1053 ± 257
HDL-C (mg/dL)	35 ± 9	39 ± 4	99 ± 17 **	90 ± 12
BUN (mg/dL)	16.0 ± 1.5	17.3 ± 1.2	68.2 ± 20.5 **	51.2 ± 26.7
Cre (mg/dL)	0.2 ± 0.1	0.2 ± 0.0	0.3 ± 0.1	0.3 ± 0.1
Serum 8-OHdG (ng/mL)	0.73 ± 0.03	0.86 ± 0.16	1.27 ± 0.35 **	0.84 ± 0.09 ##
Urine protein (mg/mg Cre)	0.6 ± 0.4	0.7 ± 0.3	5.9 ± 2.5 **	1.6 ± 0.7 ##
Urine KIM-1 (ng/mg Cre)	1.7 ± 1.2	0.8 ± 0.8	33.2 ± 9.5 **	13.4 ± 3.8 ##

Values are mean ± standard deviation. **, *p* < 0.01 compared with control rats. ##, *p* < 0.01 compared with streptozotocin-induced diabetic rats. HR: heart rate, SBP: systolic blood pressure, DBP: diastolic blood pressure, BG: blood glucose, HbA1c: glycated hemoglobin, T-Chol: total cholesterol, TG: triglycerides, HDL-C: high-density lipoprotein cholesterol, BUN: blood urea nitrogen, Cre: creatinine, 8-OHdG: 8-hydroxy-2′-deoxyguanosine, KIM-1: kidney injury molecule-1. Data were analyzed by ANOVA with Tukey–Kramer test.

## Data Availability

The datasets used and analyzed during the current study are available from the corresponding author upon reasonable request.

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
