# Peer review of "Apixaban Inhibits Progression of Experimental Diabetic Nephropathy by Blocking Advanced Glycation End Product-Receptor Axis"

_ijms, 2025, doi:10.3390/ijms26073007_

Round 1

Reviewer 1 Report

Comments and Suggestions for Authors

The present study investigated the potential therapeutic effects of apixaban against diabetic nephropathy in STZ-induced type 1 diabetic rats. The authors suggested that apixaban can ameliorate renal injury by blocking the AGE-RAGE-oxidative stress and inflammation axis in diabetic kidneys. The study is interesting, of translational value and confers great relevance to the clinical setting. However, there are major concerns that should be addressed before the manuscript is approved for publication:

  1. In the abstract-methods, line 17, the authors should indicate that type 1 diabetes was induced by streptozotocin and should mention the dose and route of administration, so that the abstract can stand independently.
  2. The instruction is poorly written and did not provide sufficient background. It should be revised in more professionalism. The authors should focus on relevant literature to diabetic nephropathy as well as the mechanism, clinical efficacy and safety profile of the studied drug (apixaban). The first paragraph is discussing arterial fibrillation, which is away from the scope of the manuscript.
  3. What is rational of apixaban selection in management of diabetic nephropathy?
  4. The novelty of the study was not clear in the introduction section. The authors should clarify the novelty of study.
  5. In Materials and methods section, line 231, what is the type and pH of the buffer used?
  6. Lines 232-234, why were diabetic rats injected with insulin? Did the authors make sure that the insulin did not significantly blood glucose levels?
  7. Why was the dosage of 5 mg/kg/BW of apixaban selected? Was it according to preliminary experiments or previous published studies? The authors should clarify the basis of choosing this dose. Additionally, the authors should mention the duration of apixaban therapy.
  8. The therapeutic effect of apixaban should be investigated in multiple doses to clarify if there is dose-response relationship.
  9. Lines 238-239, the catalogue number of 8-OHdG ELISA kit should be provided.
  10. Table 1, the title should be replaced by “ General characteristics of experimental animals or Biochemical measurements”
  11. Table 1: the results of creatinine (did not change) and serum HDL (become higher) in STZ-induced diabetic rats were inconsistent and illogic. The authors should provide reasonable explanation for these dataset.
  12. The quality of figures is poor and should be improved.
  13. Page 5, the authors should clarify that Figure 3 g is Masson’s trichrome-stained sections
  14. The sample size should be defined in each figure caption and the statistical test as well.
Comments on the Quality of English Language
  1. There are grammar, spelling and syntax mistakes throughout the manuscript that should be corrected.

Author Response

Responses to Reviewer #1

Thank you very much for your careful review and constructive comments.

Major comments:
#1. RE:
In the abstract-methods, line 17, the authors should indicate that type 1 diabetes was induced by streptozotocin and should mention the dose and route of administration, so that the abstract can stand independently.

Thank you for your comments. According to your suggestion, we added the following sentence in the revised abstract; Six-week-old Wistar rats received single 60 mg/kg intraperitoneal injection of streptozotocin to produce a model of type 1 diabetes.

#2. RE: The instruction is poorly written and did not provide sufficient background. It should be revised in more professionalism. The authors should focus on relevant literature to diabetic nephropathy as well as the mechanism, clinical efficacy and safety profile of the studied drug (apixaban). The first paragraph is discussing arterial fibrillation, which is away from the scope of the manuscript.

#3. RE: What is rational of apixaban selection in management of diabetic nephropathy?

Thank you for your valuable comments. According to your comments 2 and 3, we provided sufficient background in the first and second paragraphs of introduction section, especially focusing on relevant literature to diabetic nephropathy as well as the mechanism, clinical efficacy and safety profile of the studied drug (apixaban) as follows. Diabetic nephropathy is a leading cause of end-stage renal disease and is associated with the increased risk of cardiovascular events in patients with diabetes, which have accounted for high mortality rate in these patients [1,2]. Various metabolic and hemodynamic pathways are activated under diabetic conditions; chronic hyperglycemia-induced oxidative stress generation, inflammation, and advanced glycation end products (AGEs) formation have been reported to play a central role in the development and progression of diabetic nephropathy partly by inducing thrombotic and fibrotic reactions as well as activation of renin-angiotensin-aldosterone system [3-6]. Randomized clinical trials have shown that inhibitors of sodium-glucose cotransporter 2 or renin-angiotensin system, a glucagon-like peptide-1 receptor agonist, semaglutide, and a non-steroidal mineralocorticoid receptor antagonist, finerenone significantly reduce the risk of renal composite outcome in diabetic patients [7-10]. However, there is a high residual risk of chronic kidney disease progression with current therapies, and a substantial number of diabetic patients still experience end-stage renal disease [1,2]. Therefore, a novel therapeutic target should be clarified for the treatment of diabetic nephropathy. Atrial fibrillation (AF) is the most common disorder of cardiac rhythm, and number of patients with AF is increasing all over the world, especially in an aging society [1-4]. Most of the AF patients are affected by various comorbidities, such as hypertension, diabetes, obesity, and chronic kidney disease, all of which could increase the risk of AF-associated complications, including stroke, heart failure, thromboembolic events, and death [11-20]. A systematic review of direct oral anticoagulants (DOACs) in non-valvular AF patients with moderate chronic kidney disease revealed that dabigatran 150 mg twice daily and an inhibitor of factor Xa, apixaban were superior to warfarin for stroke and systemic embolic events prevention, while apixaban and edoxaban significantly reduced the risk of major hemorrhage more than warfarin [21,22]. Moreover, apixaban was more effective than warfarin for reduced the risk of stroke and death irrespective of renal function [23]. In addition, secondary analyses of randomized clinical trials with DOACs revealed that there were no treatment interactions with regards to prevention of systemic embolic events or stroke for any of the DOACs in patients with chronic kidney disease, diabetes, or old age [24]. Since AF and chronic kidney disease patients share risk factors, such as diabetes [11-20], the above-mentioned findings suggest that apixaban may be preferable to warfarin as a therapeutic option of anticoagulants for the prophylaxis of stroke in non-valvular AF patients with diabetic nephropathy. Furthermore, given the pathological role of oxidative stress- and/or AGE-evoked hypercoagulability in diabetic nephropathy [3-6], apixaban may exert beneficial effects on diabetic nephropathy. However, clinical efficacy of apixaban in diabetic nephropathy remains to be elucidated. This is a rationale of apixaban selection in the present study for investigating the protective role of factor Xa inhibitors against diabetic nephropathy. We also cited more recent, high-impact literature in the introduction section [refs. 1-10].

#4. RE: The novelty of the study was not clear in the introduction section. The authors should clarify the novelty of study.

Thank you for your valuable comments. According to your comments, the novelty of the study was clarified in the introduction section as follows. Given the bidirectional relationship between chronic kidney disease progression and cardiovascular events, including AF and heart failure in diabetic patients [11-14,19,32], the AGE-RAGE axis could be a therapeutic target for diabetes- or aging-related disorders, such as diabetic nephropathy in patients complicated with AF. However, the effects of apixaban on AGE-RAGE axis in the diabetic kidneys remain unexplored. In this study, we examined for the first time whether apixaban could inhibit the progression of diabetic nephropathy by reducing the AGE-RAGE axis in streptozotocin-induced type 1 diabetic rats. Novelty of this study is to clarify whether and how apixaban, one of the widely used DOACs for the treatment of non-valvular AF, could inhibit the progression of diabetic nephropathy, which is commonly associated with this most prevalent cardiac arrhythmia. We added these statements in the third paragraph of introduction section of the revised manuscript.

#5. RE: In Materials and methods section, line 231, what is the type and pH of the buffer used?

Thank you for your comment. According to your comment, we provided the type and pH of the buffer used in the methods section.

#6. RE: Lines 232-234, why were diabetic rats injected with insulin? Did the authors make sure that the insulin did not significantly blood glucose levels?

Thank you for your comments. As to your comments, diabetic rats received a low-dose of insulin (1 unit of insulin, Sigma-Aldrich) to prevent ketoacidosis. We added the statement in the methods section. As shown in Table 1, we confirmed that the low-dose of insulin did not significantly reduce blood glucose levels to control levels.

#7. RE: Why was the dosage of 5 mg/kg/BW of apixaban selected? Was it according to preliminary experiments or previous published studies? The authors should clarify the basis of choosing this dose. Additionally, the authors should mention the duration of apixaban therapy.

Thank you for your comments. As to your comments, diabetic and non-diabetic rats received 5 mg/kg BW apixaban orally once a day for 8 weeks according to the preclinical data on pharmacokinetics and pharmacodynamics of apixaban [49]. We added the statement in the methods section and cited a new reference 49.

#8. RE: The therapeutic effect of apixaban should be investigated in multiple doses to clarify if there is dose-response relationship.

Thank you for your comment. As to your comment, in the present study, we did not know the dose-response effects of apixaban on experimental diabetic nephropathy. We added the statement in the last paragraph (limitations) of discussion section.

#9. RE: Lines 238-239, the catalogue number of 8-OHdG ELISA kit should be provided.

Thank you for your comment. As to your comment, we provided the catalogue number of 8-OHdG ELISA kitin the methods section.

#10. RE: Table 1, the title should be replaced by “General characteristics of experimental animals or Biochemical measurements”

Thank you for your comment. According to your comment, we changed the title of Table 1 to “General characteristics of experimental animals”.

#11. RE: Table 1: the results of creatinine (did not change) and serum HDL (become higher) in STZ-induced diabetic rats were inconsistent and illogic. The authors should provide reasonable explanation for these dataset.

Thank you for your comments. As you pointed out, there was no significant difference of creatinine levels between control and type 1 diabetic rats. Since BW was significantly lower in streptozotocin-treated diabetic rats than control rats, skeletal muscle loss in the former group may partly explain the result. We did not know the exact reason why HDL-cholesterol levels were significantly higher in type 1 diabetic rats than controls. In any case, further clinical studies are needed to clarify whether a therapeutic dose of apixaban could attenuate renal injury evaluated by urinary excretion levels of KIM-1 and protein in patients with diabetic nephropathy via the suppression of AGE-RAGE-oxidative stress axis. We added these statements in the last paragraph (limitations) of discussion section.

#12. RE: The quality of figures is poor and should be improved.

Thank you for your comment. According to your comment, quality of figures was improved.

#13. RE: Page 5, the authors should clarify that Figure 3 g is Masson’s trichrome-stained sections.

Thank you for your comment. According to your comment, we clarified that Figure 3g was Masson’s trichrome-stained sections.

#14. RE: The sample size should be defined in each figure caption and the statistical test as well.

Thank you for your comment. According to your comment, the sample size was defined in each figure caption and the statistical test as well.

Comments on the Quality of English Language

#15. RE: There are grammar, spelling and syntax mistakes throughout the manuscript that should be corrected.

Thank you for your comment. According to your comment, grammar, spelling and syntax mistakes throughout the manuscript were corrected.

Reviewer 2 Report

Comments and Suggestions for Authors

Article Review: "Apixaban inhibits progression of experimental diabetic 2 nephropathy by blocking advanced glycation end product-receptor axis" by Matsui et al.

The article addresses the effects of apixaban on glycemic and lipid parameters, including its influence on AGEs, their receptor, and oxidative stress markers. The topic is timely and aligns well with the journal’s scope.

Abstract: All abbreviations must be defined upon first use to ensure clarity for a broad readership. Furthermore, the abstract would benefit from the inclusion of p-values for key results to provide statistical context. Correlation data — particularly regarding the statement "Urine KIM-1 levels were positively correlated with AGEs and 8-OHdG in the kidneys and serum 8-OHdG levels" — should be complemented with precise correlation coefficients and significance values. Presenting such essential data upfront may enhance the article’s visibility and citation potential.

Introduction: The introduction is well-structured and appropriately referenced. However, a more detailed overview of apixaban, including its known mechanisms and relevance to the AGE-RAGE axis, would strengthen the background context. Additionally, while the authors emphasize that the effect of apixaban on the AGE-RAGE axis in diabetic nephropathy remains unexplored, the novelty and significance of the study should be underscored more explicitly — for example, by stating whether this is the first study to explore this pathway.

Materials and methods: The methodology section lacks sufficient detail to ensure reproducibility, which is a critical limitation. Ethical committee approval must be stated explicitly. The description of serum 8-OHdG measurement requires expansion, including catalog numbers and details on the absorbance reader. Additionally, inter- and intra-assay coefficients of variability should be reported for all kits used to verify assay precision.

The "Measurement of NADPH oxidase activity" section is notably brief and insufficient for replication. A more comprehensive description is necessary to clarify the procedure. Similarly, the statement "Other blood chemistry was analyzed with standard enzymatic methods as described previously" is overly vague. Given that parameters like BUN are later presented, the methods require detailed elaboration beyond "standard methods" to ensure transparency and reproducibility.

Results: The figure quality is suboptimal — the current size limits visibility and undermines data interpretation. Enlarging the figures is essential for clarity. Moreover, abbreviations such as VCAM-1 must be explained at first mention within the text, regardless of whether a list of abbreviations is provided. Consistency in abbreviation explanations is crucial for readability and adherence to academic standards.

Discussion: The discussion is well-developed and insightful. However, a more explicit reflection on the study’s limitations is warranted, along with a dedicated section outlining potential directions for future research. This would improve the scientific depth and practical value of the discussion.

Conclusions: The conclusion section, while summarizing key findings, remains too concise. Expanding this section to emphasize the study's clinical relevance and potential implications would leave the reader with a stronger, more impactful takeaway.

References: The reference list is incorrectly formatted and requires revision to align with journal guidelines. Notably, only seven sources are from 2020 or later, which is insufficient for a study addressing a rapidly evolving topic. Incorporating more recent, high-impact literature would reinforce the study’s relevance and scientific grounding.

Author Response

Responses to Reviewer #2

Thank you very much for your careful review and constructive comments.

Major comments:

#1. RE: Abstract: All abbreviations must be defined upon first use to ensure clarity for a broad readership. Furthermore, the abstract would benefit from the inclusion of p-values for key results to provide statistical context. Correlation data — particularly regarding the statement "Urine KIM-1 levels were positively correlated with AGEs and 8-OHdG in the kidneys and serum 8-OHdG levels" — should be complemented with precise correlation coefficients and significance values. Presenting such essential data upfront may enhance the article’s visibility and citation potential.

Thank you for your valuable comments. According to your comments, all abbreviations were defined upon first use. We provided p-values for key results, including precise correlation coefficients in the abstract.

#2. RE: Introduction: The introduction is well-structured and appropriately referenced. However, a more detailed overview of apixaban, including its known mechanisms and relevance to the AGE-RAGE axis, would strengthen the background context. Additionally, while the authors emphasize that the effect of apixaban on the AGE-RAGE axis in diabetic nephropathy remains unexplored, the novelty and significance of the study should be underscored more explicitly — for example, by stating whether this is the first study to explore this pathway.

Thank you for your valuable comments. According to your comments, we provided a more detailed overview of apixaban and the novelty and significance of this study in the second and third paragraphs of introduction section of revised manuscript as follows. A systematic review of direct oral anticoagulants (DOACs) in non-valvular AF patients with moderate chronic kidney disease revealed that dabigatran 150 mg twice daily and an inhibitor of factor Xa, apixaban were superior to warfarin for stroke and systemic embolic events prevention, while apixaban and edoxaban significantly reduced the risk of major hemorrhage more than warfarin [21,22]. Moreover, apixaban was more effective than warfarin for reduced the risk of stroke and death irrespective of renal function [23]. In addition, secondary analyses of randomized clinical trials with DOACs revealed that there were no treatment interactions with regards to prevention of systemic embolic events or stroke for any of the DOACs in patients with chronic kidney disease, diabetes, or old age [24]. Since AF and chronic kidney disease patients share risk factors, such as diabetes [11-20], the above-mentioned findings suggest that apixaban may be preferable to warfarin as a therapeutic option of anticoagulants for the prophylaxis of stroke in non-valvular AF patients with diabetic nephropathy. Furthermore, given the pathological role of oxidative stress- and/or AGE-evoked hypercoagulability in diabetic nephropathy [3-6], apixaban may exert beneficial effects on diabetic nephropathy. However, the effects of apixaban on AGE-RAGE axis in the diabetic kidneys remain unexplored. In this study, we examined for the first time whether apixaban could inhibit the progression of diabetic nephropathy by reducing the AGE-RAGE axis in streptozotocin-induced type 1 diabetic rats. Novelty of this study is to clarify whether and how apixaban, one of the widely used DOACs for the treatment of non-valvular AF, could inhibit the progression of diabetic nephropathy, which is commonly associated with this most prevalent cardiac arrhythmia.

#3. RE: Materials and methods: The methodology section lacks sufficient detail to ensure reproducibility, which is a critical limitation. Ethical committee approval must be stated explicitly. The description of serum 8-OHdG measurement requires expansion, including catalog numbers and details on the absorbance reader. Additionally, inter- and intra-assay coefficients of variability should be reported for all kits used to verify assay precision.

The "Measurement of NADPH oxidase activity" section is notably brief and insufficient for replication. A more comprehensive description is necessary to clarify the procedure. Similarly, the statement "Other blood chemistry was analyzed with standard enzymatic methods as described previously" is overly vague. Given that parameters like BUN are later presented, the methods require detailed elaboration beyond "standard methods" to ensure transparency and reproducibility.

Thank you for your valuable comments. According to your comments, ethical committee approval was stated explicitly. We also provided the detailed description about serum 8-OHdG and urinary KIM-1 measurements with inter- and intra-assay coefficients of variation, NADPH oxidase activity and other blood chemistry analyses in the methods section of revised manuscript.

#4. RE: Results: The figure quality is suboptimal — the current size limits visibility and undermines data interpretation. Enlarging the figures is essential for clarity. Moreover, abbreviations such as VCAM-1 must be explained at first mention within the text, regardless of whether a list of abbreviations is provided. Consistency in abbreviation explanations is crucial for readability and adherence to academic standards.

Thank you for your valuable comments. According to your comments, we provided enlarged Figures in the revised manuscript. Abbreviations, such as MCP-1, VCAM-1, PAR-1, and PAR-2 were explained at first mention within the text.

#5. RE: Discussion: The discussion is well-developed and insightful. However, a more explicit reflection on the study’s limitations is warranted, along with a dedicated section outlining potential directions for future research. This would improve the scientific depth and practical value of the discussion.

Thank you for your valuable comments. According to your comments, we provided a more explicit reflection on the study’s limitations, along with a dedicated section outlining potential directions for future research in the last paragraph of discussion section of revised manuscript as follows. There are several limitations in this study. First, we did not know the dose-response effects of apixaban on diabetic nephropathy in our animals. Second, there was no significant difference of creatinine levels between control and type 1 diabetic rats. Since BW was significantly lower in streptozotocin-treated diabetic rats than control rats, skeletal muscle loss in the former group may partly explain the result. Third, we did not know the exact reason why HDL-cholesterol levels were significantly higher in type 1 diabetic rats than controls. In any case, further clinical studies are needed to clarify whether a therapeutic dose of apixaban could attenuate renal injury evaluated by urinary excretion levels of KIM-1 and protein in patients with diabetic nephropathy via the suppression of AGE-RAGE-oxidative stress axis.

#6. RE: Conclusions: The conclusion section, while summarizing key findings, remains too concise. Expanding this section to emphasize the study's clinical relevance and potential implications would leave the reader with a stronger, more impactful takeaway.

Thank you for your valuable comments. According to your comments, we expanded the conclusion section to emphasize the study's clinical relevance and potential implications in the revised manuscript as follows.Our present findings suggest that apixaban could ameliorate renal injury in streptozotocin-induced type 1 diabetic rats partly by blocking the crosstalk between AGE-RAGE-oxidative stress axis and thrombin-PAR-1/factor Xa-PAR-2 system in the diabetic kidneys. Given the pathological involvement of hypercoagulability in diabetic nephropathy [33-35] and the association of diabetes with the increased risk of non-valvular AF [11,19,20], apixaban, one of the therapeutic options for preventing the stroke in patients with non-valvular AF, may provide a novel strategy for attenuating renal damage in diabetic patients.

#7. RE: References: The reference list is incorrectly formatted and requires revision to align with journal guidelines. Notably, only seven sources are from 2020 or later, which is insufficient for a study addressing a rapidly evolving topic. Incorporating more recent, high-impact literature would reinforce the study’s relevance and scientific grounding.

Thank you for your comments. According to your comments, the reference list was correctly formatted, and more recent, high-impact literature was cited (refs. 1-10) in the revised manuscript.

Round 2

Reviewer 2 Report

Comments and Suggestions for Authors

The authors responded to my comments in a satisfactory manner. The manuscript has been significantly improved and supplemented, especially in the Materials and Methods section. In my opinion, it can be accepted for publication in its present form.